# Fisetin Suppresses the Inflammatory Response and Oxidative Stress in Bronchial Epithelial Cells

**DOI:** 10.3390/nu14091841

**Published:** 2022-04-28

**Authors:** Shu-Ju Wu, Wen-Chung Huang, Ching-Yi Cheng, Meng-Chun Wang, Shu-Chen Cheng, Chian-Jiun Liou

**Affiliations:** 1Department of Nutrition and Health Sciences, Research Center for Chinese Herbal Medicine, Chang Gung University of Science and Technology, Taoyuan 33303, Taiwan; sjwu@mail.cgust.edu.tw; 2Aesthetic Medical Center, Department of Dermatology, Chang Gung Memorial Hospital, Linkou, Taoyuan 33303, Taiwan; 3Graduate Institute of Health Industry Technology, Research Center for Food and Cosmetic Safety, Chang Gung University of Science and Technology, Taoyuan 33303, Taiwan; wchuang@mail.cgust.edu.tw (W.-C.H.); jennycheng@mail.cgust.edu.tw (C.-Y.C.); 4Division of Allergy, Asthma, and Rheumatology, Department of Pediatrics, Chang Gung Memorial Hospital, Linkou, Guishan Dist., Taoyuan 33303, Taiwan; 5Department of Pediatrics, New Taipei Municipal TuCheng Hospital (Built and Operated by Chang Gung Medical Foundation), New Taipei 23656, Taiwan; 6Department of Pulmonary Infection and Immunology, Chang Gung Memorial Hospital, Linkou, Taoyuan 33303, Taiwan; 7Department of Traditional Chinese Medicine, Chang Gung Memorial Hospital, Taoyuan 33303, Taiwan; mengchun1023@gmail.com; 8Department of Nursing, Division of Basic Medical Sciences, Research Center for Chinese Herbal Medicine, Chang Gung University of Science and Technology, Taoyuan 33303, Taiwan

**Keywords:** asthma, bronchial epithelial cells, fisetin, inflammation, oxidative stress

## Abstract

Fisetin is isolated from many fruits and vegetables and has been confirmed to improve airway hyperresponsiveness in asthmatic mice. However, whether fisetin reduces inflammatory response and oxidative stress in bronchial epithelial cells is unclear. Here, BEAS-2B human bronchial epithelial cells were treated with various concentrations of fisetin and then stimulated with tumor necrosis factor-α (TNF-α) or TNF-α/interleukin-4. In addition, ovalbumin-sensitized mice were treated with fisetin to detect inflammatory mediators and oxidative stress expression. Fisetin significantly reduced the levels of inflammatory cytokines and chemokines in TNF-α-stimulated BEAS-2B cells. Fisetin also attenuated intercellular adhesion molecule-1 expression in TNF-α-stimulated BEAS-2B cells, suppressing THP-1 monocyte adhesion. Furthermore, fisetin significantly suppressed airway hyperresponsiveness in the lungs and decreased eosinophil numbers in the bronchoalveolar lavage fluid of asthmatic mice. Fisetin decreased cyclooxygenase-2 expression, promoted glutathione levels, and decreased malondialdehyde levels in the lungs of asthmatic mice. Our findings indicate that fisetin is a potential immunomodulator that can improve the pathological features of asthma by decreasing oxidative stress and inflammation.

## 1. Introduction

Asthma is a common respiratory inflammation disease. Air pollution is often a huge threat to respiratory health and increases the incidence of asthma around the world [1]. An estimated 350 million people will suffer from asthma worldwide in 2025, and asthma has become a prevalent respiratory health problem [2]. An acute asthma attack causes tracheal constriction and difficulty breathing. Inflammatory cytokines or allergens can induce the activation of tracheal epithelial cells, which not only leads to airway inflammation, but these cells then secrete more mucus to obstruct the respiratory tract [3]. Tracheal epithelial cells have an important physical barrier function and can secrete mucus and remove dust and microorganisms from the respiratory tract [4]. However, in the lungs of patients with asthma, immune cells result in the secretion of high amounts of tumor necrosis factor-α (TNF-α) to induce tracheal epithelial cells to release excessive chemokines and cytokines, which not only exacerbate and worsen the inflammatory response in the lungs but also cause remodeling of the airways [5]. Inflammatory cytokines also activate the mitogen-activated protein kinase (MAPK) and nuclear factor kappa B (NF-κB) pathways to contribute to secretion of inflammatory mediators in airways [6]. T helper 2 (Th2) cells can also release more interleukin (IL)-4 to cause the tracheal epithelial cells to release eotaxins, which attract more eosinophils into the lungs, resulting in a more severe inflammation and oxidation response and reducing the physiological function of the airways [7].

Goblet cells are epithelial cells that secrete mucus in the respiratory tract [8]. In patients with asthma, allergens or other irritants could stimulate the abnormal proliferation of goblet cells and secrete more mucus, causing breathing difficulties. Mucin glycoprotein is the main component of mucus, and the mucins released by tracheal epithelial cells mainly include MUC5AC and MUC5B [9]. Many studies have confirmed that a variety of environmental factors stimulate bronchial MUC5AC expression, including air pollutants, cigarette smoke, and microorganisms [10]. Previous studies have also found that inflammatory cytokines and environmental factors stimulate oxidative stress in tracheal epithelial cells and cause toxicity to epithelial cells, leading to cell injury [11]. In addition, reactive oxygen species (ROS) are a byproduct of the oxidative phosphorylation of the electron transport chain in mitochondria. Normally, ROS can be eliminated by antioxidants, such as superoxide dismutase, catalase, and glutathione (GSH) [12]. However, in the lungs of patients with asthma, the oxidative stress increases, leading to ROS overexpression and more cell damage in the lungs [13].

Fisetin is a flavonoid that is widely distributed in many kinds of vegetables and fruits [14]. Previous studies have found that fisetin has the ability to inhibit inflammation, suppress tumor cell proliferation, and promote antioxidant effects [15]. Previous studies have shown that fisetin significantly reduces pulmonary inflammatory responses and oxidative stress by promoting heme oxygenase-1 (HO-1) and inhibiting nitric oxide synthase expression [16]. Fisetin has also been found to attenuate airway inflammation by blocking the NF-κB signaling pathway in asthmatic mice [17]. However, how fisetin regulates the molecular mechanism of the inflammatory response and oxidative stress in bronchial epithelial cells is unclear. Hence, we would investigate whether fisetin attenuates oxidative responses and inflammatory cytokine expression in human tracheal epithelial cells. We also evaluated whether fisetin improved the molecular mechanisms underlying airway inflammatory, oxidative stress, and pathological lesions in the lungs of asthmatic mice.

## 2. Materials and Methods

### 2.1. Cell Culture

Human bronchial epithelial BEAS-2B cells were purchased from the American Type Culture Collection (ATCC, Manassas, VA, USA) and maintained in DMEM/F12 medium (Thermo Fisher Scientific, Grand Island, NY, USA) supplemented with 10% heat-inactivated fetal bovine serum (Thermo Fisher Scientific) and glutamine, penicillin, and streptomycin mixed solution (Thermo Fisher Scientific). Human monocyte THP-1 cells were obtained from the Bioresource Collection and Research Center (BCRC, Hsinchu, Taiwan) and cultured in RPMI 1640 medium (Thermo Fisher Scientific). All cells were grown in constant humidity in a 5% CO_2_ incubator at 37 °C.

### 2.2. Cytotoxicity Assay

Fisetin was purchased from Sigma–Aldrich (≥98% purity by HPLC; Sigma-Aldrich, St. Louis, MO, USA). Cell viability was detected using the cell counting kit-8 assay kit (CCK-8, Sigma) as described previously [18]. Briefly, BEAS-2B cells were seeded in a 96-well culture plate at a density of 1 × 10^3^ cells/well. The cells were treated with 0–100 μM fisetin for 24 h. Next, CCK-8 detection solution was added to the culture plate and incubated at 37 °C for 4 h. Cell viability was detected using a multimode reader (Thermo, Waltham, MA, USA) based on the absorbance at 450 nm.

### 2.3. BEAS-2B Cell Culture and Fisetin Treatment

BEAS-2B cells (2 × 10^5^ cells/mL) were seeded into 24-well culture plates in DMEM/F12 medium. BEAS-2B cells treated with 0–30 μM fisetin at 37 °C for 1 h and then incubated with 10 ng/mL TNF-α or 10 ng/mL TNF-α/IL-4 at 37 °C for 24 h. The supernatants were collected, and chemokine or cytokine levels were detected using specific ELISA kits.

### 2.4. Cell Adhesion Experiment

BEAS-2B cells (2 × 10^5^ cells/mL) were pretreated with 0–30 μM fisetin and incubated with 10 ng/mL TNF-α at 37 °C for 24 h as described previously [19]. THP-1 cells were incubated with calcein AM solution (Sigma) and co-cultured with BEAS-2B cells. Attached THP-1 cells were observed and counted using fluorescence microscopy (Olympus, Tokyo, Japan).

### 2.5. ROS Expression

ROS production was detected using 2′,7′-dichlorofluorescin diacetate (DCFH-DA) (Sigma) as described previously [19]. BEAS-2B cells (2 × 10^5^ cells/mL) were treated with 0–30 μM fisetin and stimulated with 10 ng/mL TNF-α for 24 h. The cells were then incubated with DCFH-DA and then lysed to detect ROS levels using the multimode microplate reader (BioTek SynergyHT, Bedfordshire, UK). Intracellular ROS production was assayed using fluorescence microscopy (Olympus).

### 2.6. Animal Experiments

All protocols for the animal experiments were approved by the Laboratory Animal Care Committee of Chang Gung University of Science and Technology (IACUC approval number: 2018-003). Six-week-old female BALB/c mice were purchased from the National Laboratory Animal Center (Taipei, Taiwan) and maintained in temperature-controlled animal housing. Thirty-two mice were stochastically divided into four groups (*n* = 8 in each group): normal group, mice sensitized with normal saline and treated with DMSO solution via intraperitoneal injection; ovalbumin (OVA, Sigma) group, mice sensitized with OVA and treated with DMSO solution via intraperitoneal injection; and the fisetin groups, OVA-sensitized mice treated with 5 mg/kg or 10 mg/kg fisetin (Fis 5 and Fis 10 groups, respectively) via intraperitoneal injection.

### 2.7. Mouse Sensitization and Administration of Fisetin

The process of sensitizing mice was described previously [20]. Briefly, mice were injected intraperitoneally with 50 μg OVA mixed with 0.8 mg aluminum hydroxide (Thermo, Rockford, IL, USA) in 200 μL normal saline on days 1 to 3. Inhalation of 2% OVA solution atomized vapor was used to challenge allergic reactions of the airways and lungs on days 14, 17, 20, 23, and 27 using an ultrasonic nebulizer. The mice were treated intraperitoneally with fisetin or DMSO solution 1 h before the OVA challenge or methacholine (Sigma) inhalation (on day 28). Lastly, AHR was detected on day 28, and the mice were sacrificed on day 29 to evaluate oxidative stress and the airway inflammatory response.

### 2.8. Airway Hyperresponsiveness

On day 28, 24 h after the last OVA challenge, AHR was detected to evaluate the airway function as described previously [21]. Mice inhaled aerosolized methacholine (0 to 40 mg/mL) to detect the enhanced pause (Penh) and evaluate AHR by whole-body plethysmograph (Buxco Electronics, Troy, NY, USA).

### 2.9. Bronchoalveolar Lavage Fluid

Mice were placed in an anesthesia box with 4% isoflurane to induce adequate anesthesia and euthanasia. Bronchoalveolar lavage fluid (BALF) was collected as described previously [22]. Briefly, an indwelling needle intubated the trachea, and 1 mL sterile normal saline was used to wash the lungs and airways. Subsequently, the solution was collected as BALF. The BALF was centrifuged (Cytospin 4, Thermo) and BALF cells stained with Giemsa stain solution (Sigma) to distinguish the cell morphology and count cells.

### 2.10. Immunohistochemistry

Lung tissues were fixed with 10% formalin and embedded into paraffin to cut into 6-μm sections. The sections were incubated with cyclooxygenase-2 (COX-2) antibody (1:100) (Cell Signaling Technology, Danvers, MA, USA) overnight and then treated with secondary antibody for 1 h at room temperature. Finally, DAB substrate solution was added to detect COX-2 expression as described previously [23].

### 2.11. Glutathione Assay

We added 5-sulfosalicylic acid solution to lung tissues and homogenized them using a homogenizer (FastPrep-24, MP Biomedicals, Santa Ana, CA, USA). The Glutathione Assay Kit (Sigma) was used to detect the levels of GSH in lung tissues according to the manufacturer’s instructions as described previously [19]. GSH levels were detected using a microplate reader (Thermo) at an OD of 412 nm.

### 2.12. Malondialdehyde Activity

Lung tissues were homogenized and treated with perchloric acid for protein precipitation. Malondialdehyde (MDA) activity was detected using a lipid peroxidation assay kit (Sigma) as described previously [19]. MDA activity was detected using a multi-mode microplate reader (BioTek synergy HT).

### 2.13. Real-Time PCR

BEAS-2B cells (2 × 10^5^ cells/mL) were treated with fisetin (0–30 μM) and then incubated with 10 ng/mL TNF-α at 37 °C for 4 h. RNA was extracted from mouse lung tissues and TNF-α- stimulated BEAS-2B cells using Trizol reagent (Invitrogen, Paisley, Scotland) as described previously [24]. Next, cDNA was synthesized using the cDNA synthesis kit (Bio-Rad, San Francisco, CA, USA) and specific gene expression detected using a singleplex Sybr Green system (Bio-Rad). The experimental conditions for gene amplification were pre-incubation at 95 °C for 10 min, followed by gene amplification in 40 cycles of 95 °C for 15 s and 60 °C for 1 min using a real-time spectrofluorometric thermal cycler (iCycler; Bio-Rad).

### 2.14. Western Blot Analysis

BEAS-2B cells (2 × 105 cells/mL) were treated with fisetin (0–30 μM) at 37 °C for 1 h, and then stimulated with 10 ng/mL TNF-α at 37 °C for 24 h. BEAS-2B cells were also treated with fisetin and then stimulated with 10 ng/mL TNF-α for 30 min to detect protein phosphorylation. Proteins were extracted from lung tissue and BEAS-2B cells using RIPA buffer solution, and nuclear proteins were extracted using nuclear and cytoplasmic extraction reagents (Thermo Scientific) as described previously [19]. Next, the proteins were separated on SDS acrylamide gels by electrophoresis and transferred to polyvinylidene fluoride (PVDF) membranes. The PVDF membranes were blocked in 5% nonfat skim milk and incubated with specific primary antibodies, including COX-2, p38, extracellular regulated protein kinases (ERK)1/2, ICAM-1, c-Jun N-terminal kinase (JNK), phosphorylated-p38, phosphorylated-ERK1/2, phosphorylated-JNK (Cell Signaling Technology, MA, USA), NF-kappa-B inhibitor alpha (IκB-α), lamin B1, Nuclear factor erythroid 2-related factor 2 (Nrf2), HO-1, phosphorylated-IκB-α (Santa Cruz, CA, USA), and β-actin (Sigma). The membranes were incubated with secondary antibodies as appropriate. Finally, the membranes were treated with Luminol/Enhancer solution, and specific protein signals were detected using the BioSpectrum 600 system (UVP, Upland, CA, USA).

### 2.15. ELISA

BEAS-2B cells (2 × 10^5^ cells/mL) were treated with 0–30 μM fisetin at 37 °C for 1 h and then incubated with 10 ng/mL TNF-α or 10 ng/mL TNF-α/IL-4 at 37 °C for 24 h. The supernatants were collected to detect the levels of intercellular adhesion molecule-1 (ICAM-1), IL-8, IL-6, MCP-1, and c-c motif chemokine ligand CCL5, CCL11, and CCL24 using sandwich ELISA kits (R&D, Minneapolis, MN, USA). Specific protein levels were detected using a microplate reader (Thermo) at an OD of 450 nm as described previously [25]. Fisetin (10 μM) was combined with MAPK specific inhibitors, including 10 μM SP600125 (JNK inhibitor), 10 μM SB203580 (p38 inhibitor), and 10 μM PD98059 (ERK inhibitor) (Enzo Life Sciences, Inc., Farmingdale, NY, USA), to detect the ICAM-1 protein level by ELISA.

### 2.16. Transfection and Luciferase Assays

BEAS-2B cells (2 × 10^5^ cells/mL) transfected with pNFκB-Luc plasmids (Stratagene, La Jolla, CA, USA), and cells treated with 10 μM fisetin and 10 μM MAPK specific inhibitors (Enzo Life Sciences) for 1 h, and TNF-α was stimulated for 4 h. The luciferase activity measured using the multimode microplate reader (BioTek SynergyHT, Winooski, VT, USA).

### 2.17. Data Analysis

The experimental results were analyzed by one-way analysis of variance (ANOVA) and a Dunnett post-hoc test using the SPSS statistical software package version 19.0 (SPSS Inc., Chicago, IL, USA). All results were examined based on at least three independent experiments. A *p* value of <0.05 was considered significant.

## 3. Results

### 3.1. Fisetin Reduced Pro-Inflammatory Cytokine Expressions in BEAS-2B Cells

The cytotoxicity of fisetin was detected using the CCK8 assay in BEAS-2B cells. Fisetin did not present significantly cytotoxic effects at a concentration ≤30 μM, and subsequent cell experiments used fisetin at 0–30 μM (Figure 1A). BEAS-2B cells were pretreated with or without fisetin and incubated with 10 ng/mL TNF-α at 37 °C for 24 h. Tracheal epithelial cells are an important barrier to microorganisms or air pollution particles invading the airways [5]. Inflammatory cytokines can stimulate tracheal epithelial cells to release inflammatory cytokines or chemokines to cause inflammation and oxidative damage of the lung and airways in asthma [4]. Our result demonstrated that TNF-α-stimulated BEAS-2B cells could significantly promote CCL5, MCP-1, IL-8, and IL-6 expression compared to vehicle-treated BEAS-2B cells. Compared with TNF-α-stimulated BEAS-2B cells, fisetin significantly reduced CCL5, MCP-1, IL-8, and IL-6 expression (Figure 1B–E). Fisetin could also significantly attenuate IL-6, CCL5, IL-8, and MCP-1 gene expression (Figure 1F–I). Hence, fisetin could reduce inflammation in TNF-α–stimulated BEAS-2B cells. Furthermore, eosinophil migration requires eotaxins (CCL11 and CCL24) [7]. IL-4/TNF-α–stimulated BEAS-2B cells could significantly promote CCL11 and CCL24 concentrations compared to vehicle-treated BEAS-2B cells. Fisetin significantly suppressed CCL11 and CCL24 concentrations compared to IL-4/TNF-α–stimulated cells (Figure 2A,B). Furthermore, fisetin effectively inhibited CCL11 and CCL24 gene expression (Figure 2C,D). Therefore, fisetin could significantly suppress eotaxin secretion in IL-4/TNF-α–stimulated BEAS-2B cells.

### 3.2. Fisetin Attenuated ICAM-1 and MUC5AC Expression

Inflamed tracheal epithelial cells express the adhesion factor ICAM-1 to attach more immune cells. Inflamed tracheal epithelial cells also secrete more mucus to prevent foreign molecules or microorganisms for invading the airways [10]. Our result demonstrated that TNF-α-stimulated BEAS-2B cells could significantly promote the gene expression of MUC5AC and ICAM-1 compared to vehicle-treated BEAS-2B cells. Fisetin had effectively attenuated gene expression of MUC5AC and ICAM-1 compared to TNF-α-stimulated BEAS-2B cells (Figure 3A,B). Fisetin also inhibited ICAM-1 expression compared to TNF-α-stimulated BEAS-2B cells (Figure 3C–E). Furthermore, fisetin suppressed THP-1 monocyte adhesion to BEAS-2B cell (Figure 3F,G). Our results demonstrated that fisetin could decrease MUC5AC and ICAM-1 expression for reduced invading microbes in respiratory system.

### 3.3. Fisetin Regulated NF-κB and MAPK Pathways in BEAS-2B Cells

Inflamed tracheal epithelial cells could induce the expressions of the NF-κB and MAPK pathways [11]. In TNF-α-unstimulated BEAS-2B cells, NF-κB was suppressed by IκB in the cytoplasm. TNF-α stimulation induced IκB phosphorylation to release the NF-κB heterodimer, which is able to translocate to the nucleus to express inflammatory-associated genes [6]. Compared to TNF-α-stimulated BEAS-2B cells, fisetin attenuated IκB-α protein degradation and decreased IκB-α phosphorylation (Figure 4A,B). TNF-α-stimulated BEAS-2B cells promoted p65 expression in the nucleus, and fisetin relatively decreased p65 translocation into nucleus (Figure 4A,B). MAPKs have been reported to promote inflammatory cytokine and COX-2 expression in-TNF-α-stimulated BEAS-2B cells [6]. Therefore, TNF-α-stimulated BEAS-2B cells could significantly promote the phosphorylation of p38, JNK, and ERK1/2 compared to vehicle-treated BEAS-2B cells. Interestingly, fisetin attenuated the phosphorylation of p38, JNK, and ERK1/2 compared to TNF-α-stimulated BEAS-2B cells (Figure 5). Furthermore, MAPK inhibitors combined with fisetin to detect ICAM-1 expression in cell culture medium by ELISA. To determine the cytotoxicity of MAPK inhibitors in BEAS-2B cells, the CCK8 assay was used to measure cell viability. Our result demonstrated that MAPK inhibitors did not significantly affect cell viability at concentrations ≤10 μM (data not shown). Thus, TNF-α-stimulated BEAS-2B cells treated with fisetin combined with 10 μM MAPK inhibitors could more effectively inhibit ICAM-1 concentrations compared to TNF-α-stimulated BEAS-2B cells treated with fisetin (Figure 6A). Interestingly, fisetin combined with MAPK inhibitors significantly suppressed THP-1 cell adhesion to TNF-α-stimulated BEAS-2B cells (Figure 6B). Furthermore, our results showed that fisetin combined with MAPK inhibitors could more effectively suppress luciferase activity compared to activated cells treated with fisetin (Figure 6C). Hence, our results demonstrated that fisetin effectively regulated NF-κB and MAPK pathways in TNF-α-stimulated BEAS-2B cells.

### 3.4. Effect of Fisetin on ROS Expression

Inflammation also induced oxidative stress to release more free radicals for the cell and tissue damage caused in the respiratory tract [13]. BEAS-2B cells were incubated with DCFH-DA to detect ROS levels. Our result demonstrated that TNF-α-stimulated BEAS-2B cells could significantly promote ROS expression compared to vehicle-treated BEAS-2B cells. Fisetin could effectively contribute to weakening the ROS production (Figure 7A). Using fluorescence microscopy, we observed the ROS levels, demonstrating that fisetin inhibited intracellular ROS expression (Figure 7B,C). Fisetin also enhanced the cytoplasm level of HO-1 protein and promoted the nuclear level of Nrf2 protein compared to TNF-α-stimulated BEAS-2B cells (Figure 7D,E). Here, we thought that fisetin could reduce ROS expression in TNF-α-stimulated BEAS-2B cells.

### 3.5. Fisetin Effects on AHR in Asthmatic Mice

AHR value was used to evaluate the airway function [3]. At 40 mg/mL of inhaled methacholine, the Penh values were higher in the OVA group than in normal mice. In addition, the fisetin groups had significantly reduced Penh values compared to the OVA group (Figure 8A). Hence, fisetin has a significant ability to reduced AHR in asthmatic mice.

### 3.6. Fisetin Suppressed Eosinophil Numbers in the BALF

Compared with the Normal group mice, asthmatic mice also had significantly increased numbers of eosinophils and an increased total number of cells in BALF. In the BALF of asthmatic mice treated with fisetin, the numbers of eosinophils and neutrophils significantly suppressed compared to OVA-sensitized mice. Fisetin also reduced the total number of cells in the BALF compared to the OVA group (Figure 8B).

### 3.7. Fisetin Reduced OVA-Induced Inflammatory Mediators in the Lungs

Immunohistochemical staining and Western blot of COX-2 in lung indicated that asthmatic mice treated with fisetin inhibited COX-2 production (Figure 9A–C). Moreover, fisetin effectively decreased ICAM-1 protein expression in lung compared to the OVA group (Figure 9D,E). Fisetin also effectively inhibited *TNF-α*, *IL-6*, *CCL11*, and *CCL24* gene expression (Figure 10A–D). Our results demonstrated that fisetin effectively reduced inflammation in the lungs of OVA-sensitized asthma mice.

### 3.8. Fisetin Modulated MDA and GSH Levels in the Lungs

Oxidative stress could cause lung cell damage in patients with asthma [11]. Compared with the Normal group mice, asthmatic mice also had significantly reduced GSH levels and increased MDA levels in lung tissue. Asthmatic mice treated with fisetin had significantly promoted GSH levels and inhibited MDA levels compared to OVA-sensitized mice (Figure 11A,B). Fisetin also enhanced HO-1 expression and promoted nuclear Nrf2 production in the lungs (Figure 11C,D). Thus, fisetin significantly attenuated oxidative stress in the lungs of OVA-sensitized asthma mice.

## 4. Discussion

Asthma typically presents with shortness of breath, wheezing, cough, and chest tightness [2]. Acute asthma attacks are often triggered by, among other factors, allergens, cold air, exercise, respiratory virus infection, and airborne particles [26]. Previous studies have consistently confirmed that fisetin can inhibit the AHR in asthmatic mice and inhibit Th2-related cytokine expression in the lungs [17]. Fisetin reduced airway inflammation in OVA-induced mice through the blocked MyD88/NF-κB pathway [27]. Recent studies have found that fisetin could alleviate acute and chronic asthma symptoms in asthmatic mice via regulated Th2 cell activation and suppressed airway inflammation [28,29]. In our previous study, fisetin was shown to effectively reduce the secretion of TNF-α and IL-6 in IL-1β-stimulated human lung epithelial cells [30]. Previous studies have found that fisetin promotes Nrf2 translocation into the nucleus and increases HO-1 gene expression in hepatocytes [31]. This suggests that fisetin has an anti-oxidative effect for the prevention and treatment of oxidation-stress-related diseases. However, whether fisetin can improve the expression of oxidative stress and attenuate the production of inflammatory mediators in tracheal epithelial cells and the lungs of asthmatic mice is unclear. Hence, we would investigate whether fisetin improved the molecular mechanisms of oxidative responses and inflammatory mediator expression in human tracheal epithelial cells. We also we investigated the effects of fisetin on airway inflammation, oxidative stress, and airway goblet cell proliferation in asthmatic mice.

Tracheal epithelial cells are an important barrier to microorganisms or air pollution particles invading the respiratory tract [32]. Abnormal inflammation of tracheal epithelial cells is a risk factor in the development of asthma [33]. When allergens or microorganisms invade the airways, they may stimulate the activation of immune cells in the lungs and release more inflammatory cytokines. These cytokines not only stimulate inflammation in epithelial cells but also induce the inflamed tracheal epithelial cells to release inflammatory cytokines and chemokines to exacerbate inflammation and oxidative damage of the airways [34]. This would cause damage to the tracheal epithelial cells and lungs, deteriorating lung cell function and weakening respiratory efficiency [35]. Previous studies have shown that fisetin improves asthma symptoms in asthmatic mice [27,28,29], but these studies did not investigate the importance of airway epithelial cells. In this study, we simulated the cellular model of TNF-α release by activated immune cells in the lungs affecting epithelial cell inflammation. The experimental results of this study demonstrated that fisetin effectively inhibits the secretion of inflammatory cytokines and chemokines in tracheal epithelial cells. Furthermore, fisetin effectively inhibited the adhesion of immune cells to tracheal epithelial cells, mainly through inhibiting the expression of ICAM-1. Fisetin could also improve HO-1 production and reduce the oxidative stress in inflammatory bronchial epithelial cells. Recent studies have suggested the inflamed tracheal epithelial cells release more mucus and obstruct the respiratory tract [36]. Fisetin significantly reduced the expression of *Muc5AC* and contributed to a reduction in excessive mucus secretion. Therefore, our results confirmed that fisetin effectively inhibits inflammation and oxidative stress in inflammatory tracheal epithelial cells and reduces excessive mucus secretion, reducing dyspnea in patients with asthma.

The inflammatory response in the lungs of patients with asthma is related to the activation of immune cells [37]. The activated immune cells can release TNF-α, which not only induce cell damage in the lungs but also leads to an inflammatory response in tracheal epithelial cells [38]. Thus, the inflamed tracheal epithelial cells release more inflammatory cytokines and chemokines, causing oxidative stress and cell damage to the lung cells, worsening the function of the lung cells [39]. Our experiments found that fisetin can inhibit IL-6 secretion in TNF-α-stimulated tracheal epithelial cells, which contribute to reducing inflammation in the lungs. Previous studies found that fisetin can improve the asthma symptoms of asthmatic mice and reduce TNF-α expression in the BALF [40]. Our experiments also found that fisetin can reduce the gene expression of IL-6 and TNF-α in the lungs of asthmatic mice, which indicates that fisetin has the ability to improve inflammation in the airways of asthmatic mice. Fisetin can also reduce the production of IL-8 in BEAS-2B cells activated by TNF-α. IL-8 is a chemokine that can attract neutrophils into inflamed tissues [34]. The OVA-induced asthma model mainly induced eosinophil infiltration into the lungs and can induce a small number of neutrophils to infiltrate the lungs [41]. The activated neutrophils in the lungs will release more inflammatory mediators, leading to lung inflammation and oxidative damage [42]. Our experiment found that fisetin can reduce neutrophils in BALF from asthmatic mice. Therefore, we conclude that fisetin inhibits the expression of IL-8 in bronchial epithelial cells and assist in improving the inflammatory response in the lungs of patients with asthma.

COX-2 can continue to be expressed in inflamed tissues, decomposing arachidonic acid to produce PGE2, which is an inflammatory substance [43]. In recent years, more and more studies have shown that the PGE2 signaling pathway is closely related to the development of lung diseases, which mainly include COPD, bronchial asthma, and pneumonia [44,45]. The lungs of asthmatic patients have higher levels of PGE2 secretion, which can promote the permeability of blood vessels, stimulate the expansion of microvessels, and cause more immune cells to infiltrate the lungs [43]. We found that fisetin can reduce the expression of COX-2 protein in the lungs of asthmatic mice. Using immunohistochemical staining, we also found that fisetin can effectively reduce the distribution of COX-2 in the lungs. This result shows that the reduction in COX-2 expression by fisetin contributes to reducing the expression of PGE2 in the lungs of asthmatic patients, reducing immune cell infiltration into the lungs and causing inflammation.

Higher amounts of eotaxin can be detected in the BALF or lungs of patients with asthma. Eotaxin can attract eosinophils into the lungs [34]. These eosinophils will release more inflammatory and oxidative molecules, causing lung inflammation and oxidative damage in patients with asthma [46]. Fisetin can significantly inhibit the genes expression of *CCL11* and *CCL24* in the lungs of asthmatic mice. Fisetin can also inhibit the secretion of CCL11 and CCL24 in BEAS-2B cells stimulated with TNF-α/IL-4. Therefore, fisetin will effectively reduce eosinophils in the BAFL by reducing the expression of eotaxin in asthmatic mice.

Previous studies have found that fisetin inhibits eosinophil infiltration and improves AHR in the lungs of asthmatic mice by inhibiting the NF-κB signaling pathway [17,40].

MAPK signaling also increased NF-κB activity in inflammatory calls [6]. Many previous experiments have found that the NF-κB and MAPK signaling pathways can enhance inflammatory cytokine production and increase cell adhesion molecule expression in tracheal and lung epithelial cells [30,47]. Our previous experiment found that fisetin can inhibit IL-1β-induced inflammatory cytokine and chemokine secretion in lung epithelial cells, mainly by blocking the NF-κB and MAPK signaling pathways [30]. Here, fisetin inhibited IκB phosphorylation and effectively attenuated NF-κB translocation into the nucleus in inflamed BEAS-2B cells. Fisetin also effectively attenuated phosphorylation of p38, JNK, and ERK1/2 in inflamed BEAS-2B cells. Therefore, we confirmed that fisetin effectively reduces the gene expression of *CCL5*, *MCP-1*, *IL-6*, *IL-8*, and *ICAM-1* compared to TNF-α-activated BEAS-2B cells. In addition, fisetin reduces the expression of ICAM-1 in the lungs of asthmatic mice. Recent studies have shown that the MAPK signaling pathway also induces inflammatory gene expression and ICAM-1 gene expression [48,49]. The combination of fisetin and MAPK inhibitor can inhibit the expression of ICAM-1 and reduce the adhesion of THP-1 cells to BEAS-2B cells activated by TNF-α. Therefore, fisetin can reduce the inflammatory response by inhibiting the NF-κB and MAPK signaling pathways in TNF-α-stimulated BEAS-2B cells.

Acute asthma or chronic asthma will continue to induce oxidative stress, which causes excessive mucus secretion by lung epithelial cells and tracheal epithelial cells [50,51]. In the lungs of patients with asthma, eosinophil infiltration will release eosinophil peroxidase (ECP) [52]. ECP can adhere to the cell membranes of lung cells and change their permeability. ECP can also increase the production of ROS, causing oxidative damage to lung cells [41]. Oxidative stress can stimulate bronchoconstriction in patients with asthma and increase the proliferation of bronchial epithelial cells and excessive mucus secretion, increasing airway obstruction during acute attacks and causing breathing difficulties [13]. Fisetin can inhibit MDA levels in asthmatic mice and improve the levels of GSH. Fisetin also increases the expression of Nrf2 and HO-1 in the lungs and the tracheal epithelium, improving oxidative stress. GSH has an antioxidant effect, reducing the oxidative stress that causes cell damage to the lungs and airways [12]. MDA is an important indicator of cellular oxidative stress [30]. Previous studies have found that COX-2 activation induces the expression of large amounts of PGE2, which stimulate lung cells to release more MDA [47]. Because fisetin can inhibit COX-2 expression in the lungs of asthmatic mice, it also blocks the production of MDA and reduces the oxidative damage in lung cells. Our experiments confirmed that fisetin could attenuate oxidative stress in the lungs and contributes to reducing lung damage by attenuating oxidative stress in asthmatic mice.

## 5. Conclusions

Our experimental findings confirmed that fisetin can inhibit oxidative stress in the lungs of asthmatic mice by inhibiting COX-2 expression and reducing lung inflammation. Fisetin also reduce eosinophil infiltration and AHR in asthmatic mice and can effectively inhibit the expression of the MAPK and NF-κB pathways, reducing the expression of chemokines and inflammatory cytokines in inflammatory tracheal epithelial cells. Moreover, fisetin reduces ROS levels, attenuating the oxidative damage in lung cells and bronchial epithelial cells (Figure 12). Our results provide new insights into the potential pharmacologic benefits of fisetin, including the ability to improve allergic inflammation and antioxidant effects in asthma patients.

## Figures and Tables

**Figure 1 nutrients-14-01841-f001:**
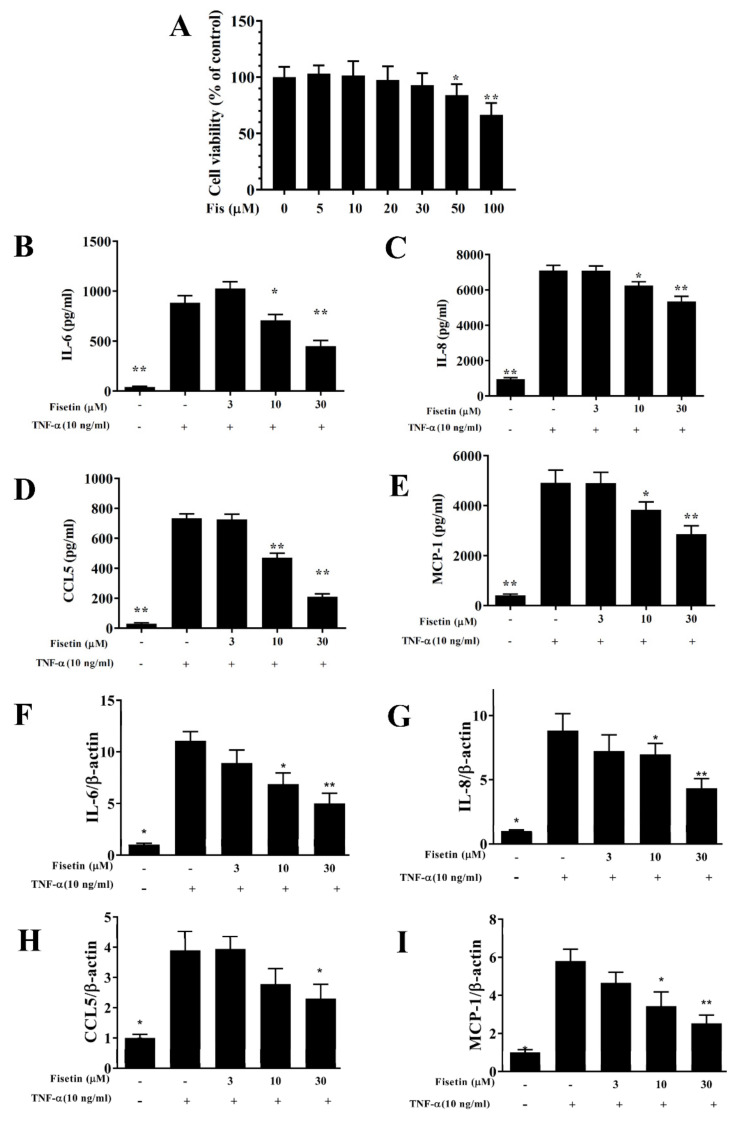
The inflammatory response was inhibited by treatment with fisetin in TNF-α-induced BEAS-2B cells. (**A**) Cell viability detected by CCK-8 reagent, and data expressed as mean ± SEM. * *p* < 0.05, ** *p* < 0.01, compared to normal BEAS-2B cells. (**B**) IL-6, (**C**) IL-8, (**D**) CCL5, and (**E**) MCP-1 expressions detected by ELISA (*n* = 12 per group). (**F**) The gene expression of IL-6, (**G**) IL-8, (**H**) CCL5, and (**I**) MCP-1 determined using real-time PCR. The fold expression presented relative to β-actin (*n* = 8 per group). The data are presented as mean ± SEM of three independent experiments. * *p* < 0.05, ** *p* < 0.01, compared to the TNF-α-stimulated BEAS-2B cell control.

**Figure 2 nutrients-14-01841-f002:**
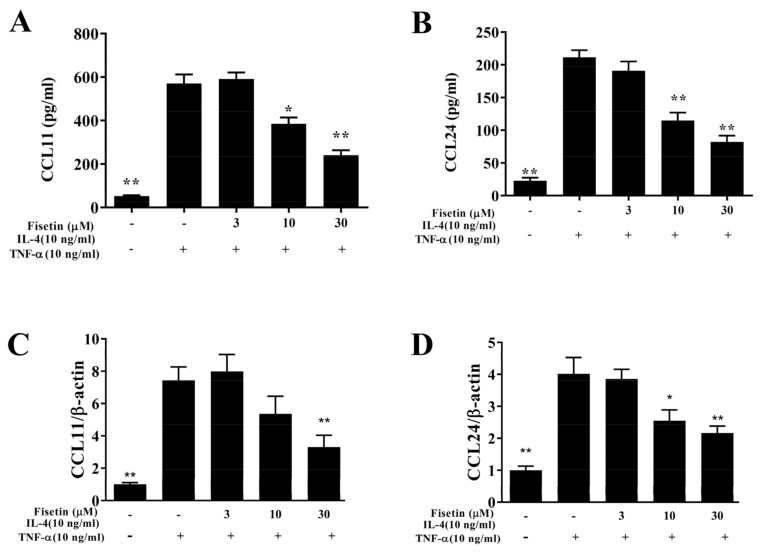
Eotaxin expression was inhibited by treatment with fisetin in TNF-α/IL-4-induced BEAS-2B cells. (**A**) CCL11 and (**B**) CCL24 detected by ELISA (*n* = 12 per group). (**C**) Gene expression of CCL11 and (**D**) CCL24 detected by real-time PCR (*n* = 8 per group). The fold expression presented relative to β-actin. The data presented as mean ± SEM of three independent experiments. * *p* < 0.05, ** *p* < 0.01, compared to the TNF-α/IL-4-stimulated BEAS-2B cell control.

**Figure 3 nutrients-14-01841-f003:**
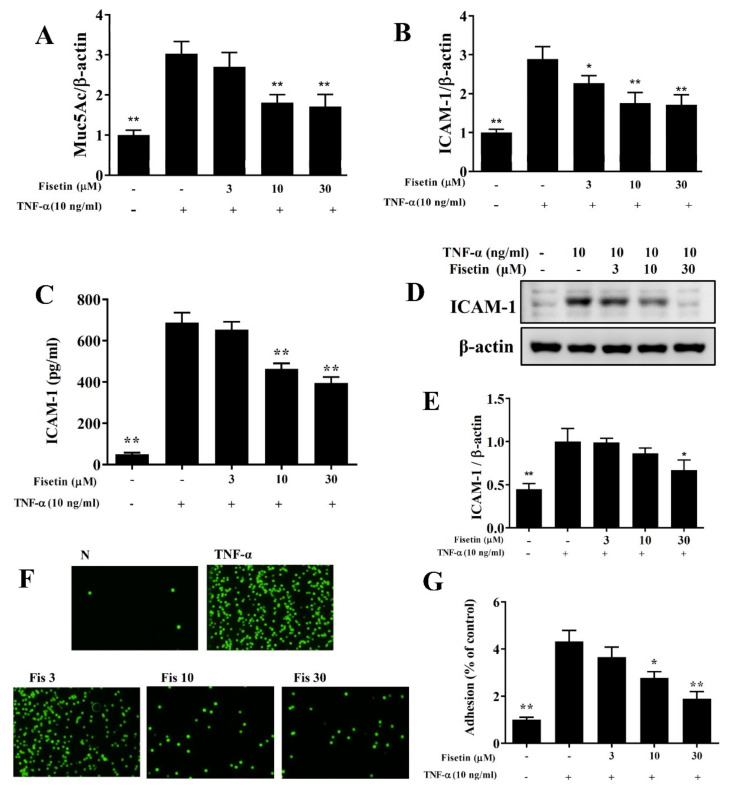
MUC5AC and ICAM-1 expression suppressed by treatment with fisetin (Fis) in TNF-α-induced BEAS-2B cells. (**A**) MUC5AC and (**B**) ICAM-1 gene expression detected by real-time PCR. The fold expression presented relative to β-actin expression (*n* = 8 per group). (**C**) ELISA (*n* = 12 per group) and (**D**) Western blot investigated ICAM-1 expression (*n* = 3 per group). (**E**) The fold expression relative to β-actin. (**F**) THP-1 cells (fluorescently labeled green) co-cultured with unstimulated (N) or TNF-α-stimulated BEAS-2B cells treated with Fis. (**G**) Percentage of THP-1 cell adhesion to BEAS-2B cells. The data presented as mean ± SEM of three independent experiments. * *p* < 0.05, ** *p* < 0.01, compared to the TNF-α-stimulated BEAS-2B cell control.

**Figure 4 nutrients-14-01841-f004:**
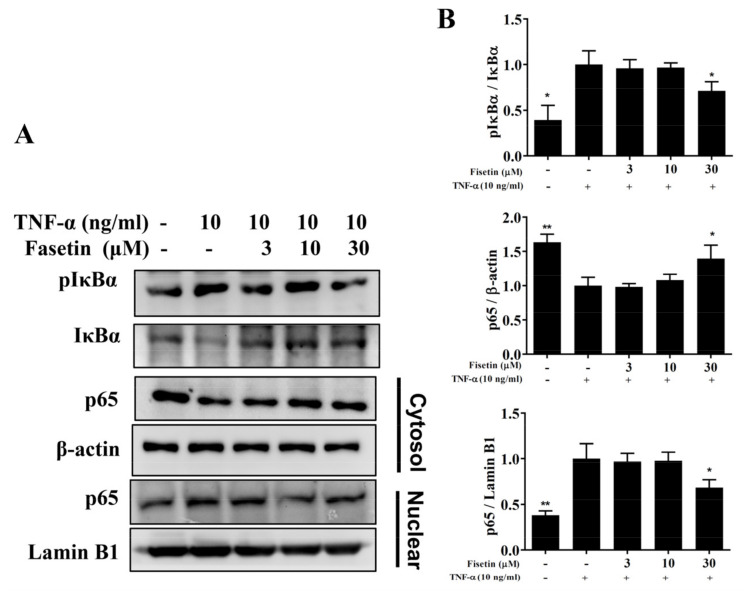
Fisetin treatment suppressed nuclear translocation of NF-κB in TNF-α-induced BEAS-2B cells. (**A**) Fisetin regulated NF-κB pathway expression (*n* = 3 per group). (**B**) The fold expression of pIκBα and p65 in the cytoplasm, and the levels of p65 in the nucleus. The data presented as mean ± SEM of three independent experiments. * *p* < 0.05, ** *p* < 0.01, compared to the TNF-α-stimulated BEAS-2B cell control.

**Figure 5 nutrients-14-01841-f005:**
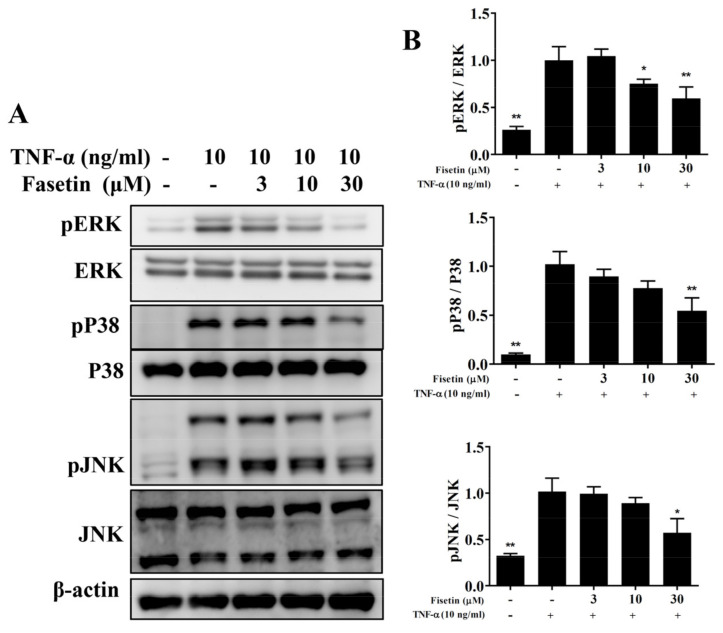
MAPK phosphorylation suppressed by treatment with fisetin in TNF-α-induced BEAS-2B cells. (**A**) Phosphorylated levels of ERK, JNK, and p38 proteins (*n* = 3 per group). (**B**) The fold expression of pERK, pP38, and pJNK measured relative to the expression of ERK, P38, and JNK, respectively. The data are presented as mean ± SEM of three independent experiments. * *p* < 0.05, ** *p* < 0.01, compared to the TNF-α-stimulated BEAS-2B cell control.

**Figure 6 nutrients-14-01841-f006:**
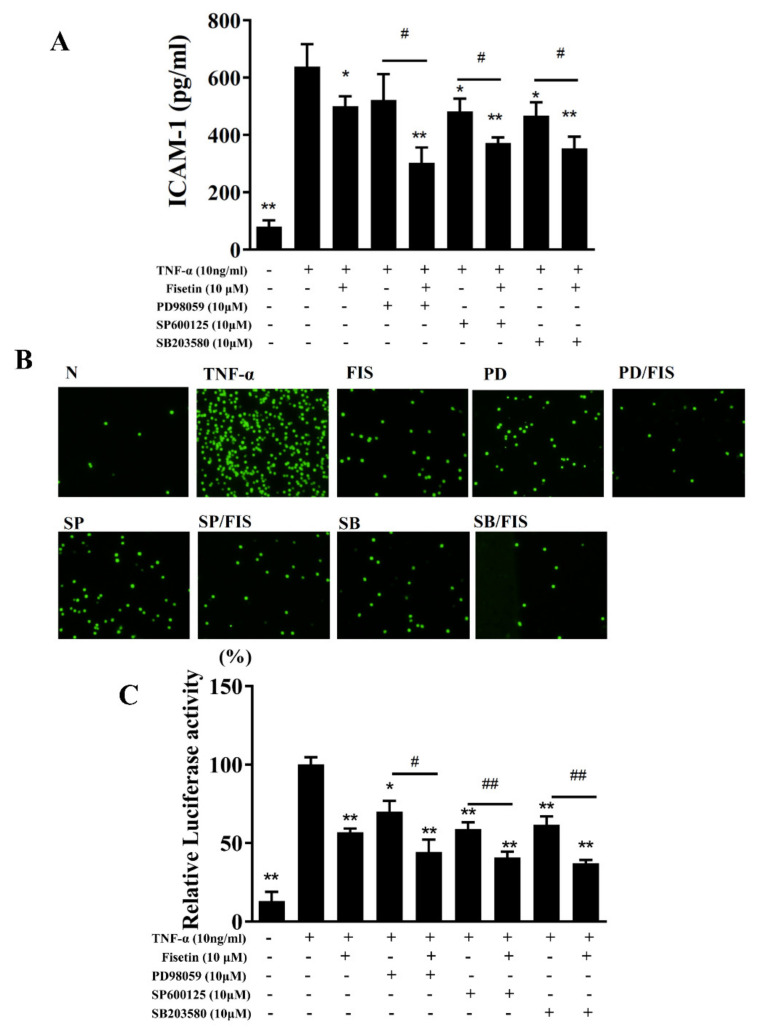
ICAM-1 expression suppressed by treatment with fisetin (Fis) and MAPK inhibitors in TNF-α-induced BEAS-2B cells. (**A**) ICAM-1 levels detected by ELISA. (**B**) THP-1 cells (fluorescently labeled green) co-cultured with unstimulated (N) or TNF-α–stimulated BEAS-2B cells treated with Fis and MAPK inhibitors. (**C**) Fisetin and MAPK inhibitors decreased luciferase activity as determined by the NF-κB promoter assay. The data are presented as mean ± SEM of three independent experiments. * *p* < 0.05, ** *p* < 0.01, compared to the TNF-α-stimulated BEAS-2B cell control. # *p* < 0.05, ## *p* < 0.01, compared to MAPK inhibitor-treated TNF-α-activated BEAS-2B cells. N: untreated cells; TNF-α: TNF-α-activated BEAS-2B cells; Fis: 10 μM fisetin treated with TNF-α-activated BEAS-2B cells; PD: PD98059 treated with TNF-α-activated BEAS-2B cells; PD/Fis: PD98059 and 10 μM fisetin treated with TNF-α-activated BEAS-2B cells; SP: SP600125 treated with TNF-α-activated BEAS-2B cells; SP/Fis: SP600125 and 10 μM fisetin treated with TNF-α-activated BEAS-2B cells; SB: SB203580 treated with TNF-α-activated BEAS-2B cells; SB/Fis: SB203580 and 10 μM fisetin treated with TNF-α-activated BEAS-2B cells.

**Figure 7 nutrients-14-01841-f007:**
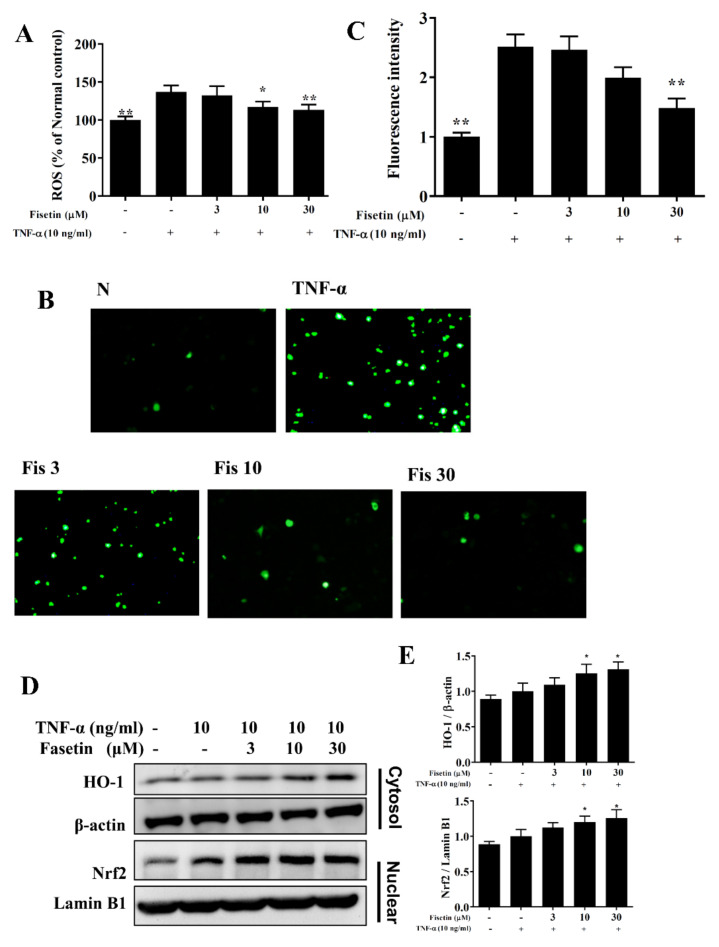
ROS production suppressed by treatment with fisetin (Fis) in TNF-α-induced BEAS-2B cells. (**A**) Percentages of ROS compared to TNF-α-activated BEAS-2B cells. (**B**) Fluorescent images of intracellular ROS. (**C**) Fluorescence intensity of intracellular ROS. (**D**) Western blot of HO-1 and Nrf2 protein expression. (**E**) The fold expression of Nrf2 and HO-1 calculated relative to lamin B1 and β-actin, respectively. The data are presented as mean ± SEM of three independent experiments. * *p* < 0.05, ** *p* < 0.01, compared to the TNF-α-stimulated BEAS-2B cell control. N: untreated cells; TNF-α: TNF-α-activated BEAS-2B cells; 3 μM, 10 μM, and 30 μM fisetin treated with TNF-α-activated BEAS-2B cells were named as Fis 3, Fis 10 and Fis 30, respectively.

**Figure 8 nutrients-14-01841-f008:**
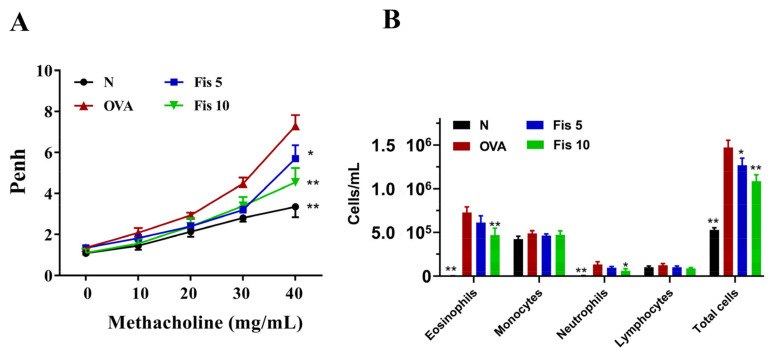
Effect of fisetin (Fis) on AHR and cell counts in BALF. (**A**) Mice inhaled increasing doses of methacholine before assessing AHR, which is shown as Penh values. (**B**) Inflammatory cells in the BALF were counted. The data presented as mean ± SEM of three independent experiments (*n* = 8 per group). * *p* < 0.05, ** *p* < 0.01, compared to the OVA control group. N: Normal group; OVA: mice sensitized with OVA. OVA-sensitized mice treated with 5 mg/kg or 10 mg/kg fisetin were named as Fis 5 and Fis 10, respectively.

**Figure 9 nutrients-14-01841-f009:**
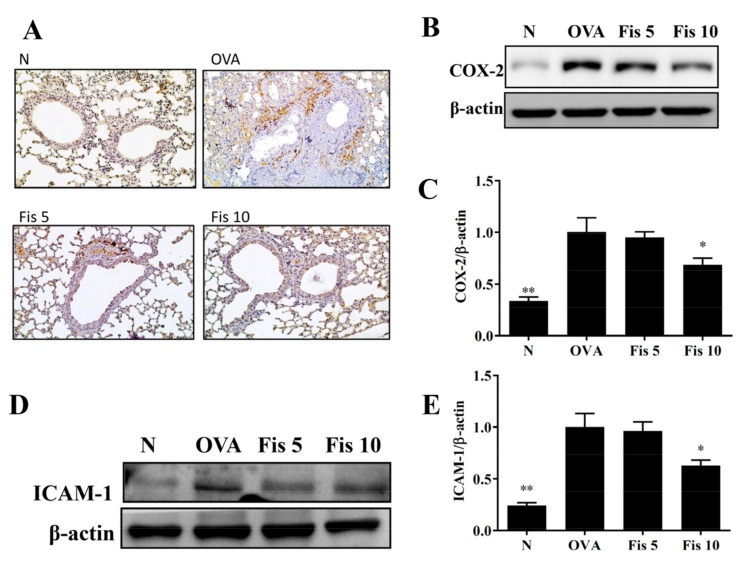
Effects of fisetin on lung tissue in asthmatic mice. Lung tissues were from normal (N) and OVA-stimulated (OVA) mice with or without fisetin (Fis 5, Fis 10) treatment. (**A**) COX-2 expression in the lungs by immunohistochemistry. (**B**) Western blots of COX-2 and (**D**) ICAM-1 expression in lung tissue. (**C**) The fold-change in COX-2 and (**E**) ICAM-1 measured relative to the expression of β-actin. The data are presented as mean ± SEM of three independent experiments, *n* = 3–5. * *p* < 0.05, ** *p* < 0.01, compared to the OVA control group. N: Normal group; OVA: mice sensitized with OVA. OVA-sensitized mice treated with 5 mg/kg or 10 mg/kg fisetin were named as Fis 5 and Fis 10, respectively.

**Figure 10 nutrients-14-01841-f010:**
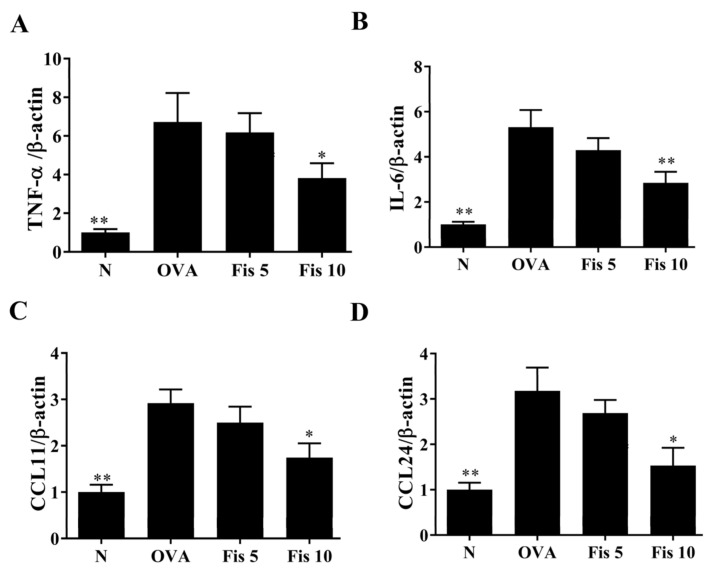
Gene expression in lung tissue from normal (N) and OVA-stimulated (OVA) mice with or without fisetin (Fis 5, Fis 10) treatment. (**A**) *TNF-α*, (**B**) *IL-6*, (**C**) *CCL11,* (**D**) *CCL24*. Fold changes in expression were measured relative to β-actin. The data are presented as mean ± SEM of three independent experiments, *n* = 6. * *p* < 0.05, ** *p* < 0.01, compared to the OVA control group. N: Normal group; OVA: mice sensitized with OVA. OVA-sensitized mice treated with 5 mg/kg or 10 mg/kg fisetin were named as Fis 5 and Fis 10, respectively.

**Figure 11 nutrients-14-01841-f011:**
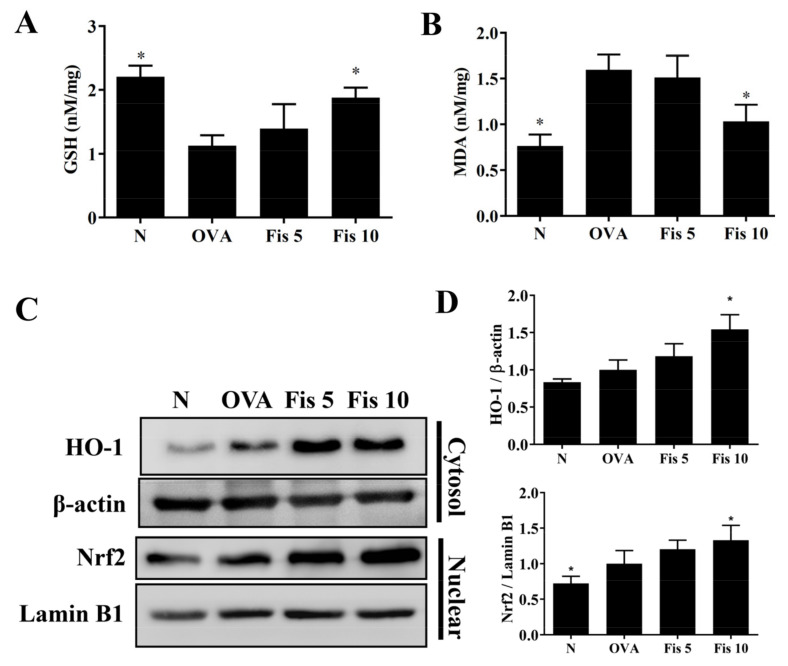
Effects of fisetin on oxidative stress factors in lung tissue. (**A**) GSH activity and (**B**) MDA activity measured in lung tissue. (**C**) Western blot of HO-1 and Nrf2 expression. (**D**) The fold expression of Nrf2 and HO-1 relative to the expression of lamin B1 and β-actin, respectively. The data are presented as mean ± SEM of three independent experiments, *n* = 3–6. * *p* < 0.05, compared to the OVA control group. N: Normal group; OVA: mice sensitized with OVA. OVA-sensitized mice treated with 5 mg/kg or 10 mg/kg fisetin were named as Fis 5 and Fis 10, respectively.

**Figure 12 nutrients-14-01841-f012:**
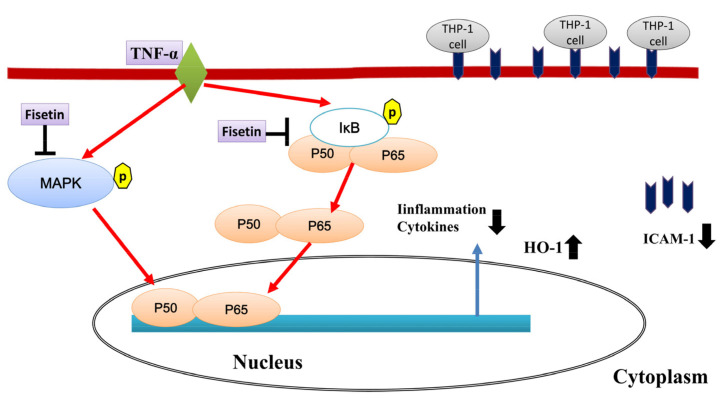
Model explaining the mechanism for the anti-inflammatory and antioxidant effects of fisetin in TNF-α-induced BEAS-2B cells.

## Data Availability

The data presented in this study are available on request from the corresponding author.

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
