# Peer review of "Fisetin Suppresses the Inflammatory Response and Oxidative Stress in Bronchial Epithelial Cells"

_nutrients, 2022, doi:10.3390/nu14091841_

Round 1
Reviewer 1 Report
This work demonstrated efficiency of fisetin in suppression of inflammation and oxidative stress using the human bronchial epithelial cell line and the OVA sensitized asthma mouse models. The results are very clear and presented excellently. I have a few comments below.
Major comments,
The authors performed the experiments using THP-1 monocytes adhesion. What does it mean immunologically “ less adhesion” to the BEAE-2B cells?
No explanation for Fig 5 in the text.
Abstract
Line 27, put comma after vegetables.
Line 28, delete period after reduces.
The authors should fully spell the abbreviation at the first appearance such as HO-1, CCK-8 TNF, IL, etc.
Line 264, The cells incubated with → BEAS-2B cells were incubated with
Line 303, asthma mice → OVA sensitized asthma mice
Author Response
This work demonstrated efficiency of fisetin in suppression of inflammation and oxidative stress using the human bronchial epithelial cell line and the OVA sensitized asthma mouse models. The results are very clear and presented excellently. I have a few comments below.
Major comments,
1.The authors performed the experiments using THP-1 monocytes adhesion. What does it mean immunologically “ less adhesion” to the BEAE-2B cells?
Responses:
Inflamed airway epithelial cells will express cellular adhesion factors, including ICAM-1, to adhere immune cells and infiltrate into the lungs. Asthma patients have a lot of eosinophilic infiltration in the lungs and respiratory tract. In addition, tracheal epithelial cells from asthmatic patients release eotaxins (CCL11 and CCL24) to attract mature eosinophil migration into the lungs. The activated eosinophils release a large amount of basic proteins, stimulate mast cell activation and degranulation, and release allergic mediators to cause allergic reactions in the lungs of asthmatic patients. Hence, THP-1 cells simulated the adherence of immune cells in the inflamed tracheal epithelial cell model. Fisetin reduced the expression of ICAM-1 in inflamed tracheal epithelial cells, reducing the adherence of THP-1 cells to tracheal epithelial cells.
2.No explanation for Fig 5 in the text.
Responses:
Thank for reviewer’s suggestion. We add some sentences to explain that fisetin regulates the MAPK signaling pathway in Discussion (Line 453-463)
Abstract
3.Line 27, put comma after vegetables.
Responses:
Thanks for your kind reminder
4.Line 28, delete period after reduces.
Responses:
Thanks for your kind reminder
5.The authors should fully spell the abbreviation at the first appearance such as HO-1, CCK-8 TNF, IL, etc.
Responses:
Thank for reviewer’s suggestion. We carefully modify the abbreviation at the first appearance in this manuscript.
6.Line 264, The cells incubated with → BEAS-2B cells were incubated with
Responses:
Thanks for your kind reminder (Line 308)
7.Line 303, asthma mice → OVA sensitized asthma mice
Responses:
Thanks for your kind reminder (Line 351)

Reviewer 2 Report
In the present manuscript, the antinflammatory and antioxidant properties of the flavonoid fisetin have been evaluated in TNF-α and IL-6-activated BEAS-2B cells and in an animal model of ovalbumin induced airway inflammation. The beneficial effects of the substance have been found mediated by the modulation of different pathways involved in both inflammation and oxidative stress, including MAPK, NF-κB and Nrf2 cascades. Moreover, fisetin was able to suppress the immune response activated by inflammatory stimuli. The obtained results agree with previous published data, which highlighted a similar power of fisetin in different model of airway inflammation (see for instance Huang W, Li ML, Xia MY, Shao JY. Fisetin-treatment alleviates airway inflammation through inhbition of MyD88/NF-κB signaling pathway. Int J Mol Med. 2018 Jul;42(1):208-218. doi: 10.3892/ijmm.2018.3582; Paul, P., Majhi, S., Mitra, S., & Banerjee, E. R. (2019). Orally administered fisetin as an immuno-modulatory and therapeutic agent in a mouse model of chronic allergic airway disease. Biomedical Research and Therapy, 6(7), 3262-3273. https://doi.org/10.15419/bmrat.v6i7.553; Wu MY, Hung SK, Fu SL. Immunosuppressive effects of fisetin in ovalbumin-induced asthma through inhibition of NF-κB activity. J Agric Food Chem. 2011 Oct 12;59(19):10496-504. doi: 10.1021/jf202756f; Tanaka, T.; Takahashi, R. Flavonoids and Asthma. Nutrients 2013, 5, 2128-2143. https://doi.org/10.3390/nu5062128; Zhang H, Zheng W, Feng X, et al. Nrf2⁻ARE Signaling Acts as Master Pathway for the Cellular Antioxidant Activity of Fisetin. Molecules. 2019;24(4):708. doi:10.3390/molecules24040708). However, the true novelty of the study and its contribution to the characterization of fisetin bioactivities and pharmacological interest remain to be defined.
This represents a major limit of the study along with reading difficulties caused by a lack of clarity in the English style and contents. This last issue requires a deep and entire revision of the manuscript, likely taking advantage of the advice of English mother tongue.
Other issues to be considered are described following.
Lines 78-81: The aim of the study should be better described, also explaining the study design and the choice of the applied experimental models.
- Cell culture studies: the choice of the concentrations used along with time exposure should be explained.
- Line 104: the number of seeded cells should be reported
- Animal experiments: the choice of administered dose, administration route, methods for euthanasia should be clarified; moreover, equipment used to allow the reagent inhalation should be provided
- Paragraphs 2.12, 2.13, 2.14: the number of seeded cells should be reported.
- Line 165: the choice inhibitor concentrations should be explained
- 16. Data analysis: “three experiments including” and how many technical replicates?
- Figures should be improved, using the same font. Font must be nonitalicized
- Figure captions: the number of biological and technical replicates should be reported
- Figure 2 should be reported before the paragraph 3.2
- Significance should be evaluated with respect to the vehicle and the pro-inflammatory stimulus and symbols should be reported in the figures accordingly.
- Lines 308-309 “Thus, fisetin im-308 proved significantly oxidative stress in OVA sensitized mice”: the wording is confusing
- Methods and Results should be rewritten in a more clear and fluent form
- Discussion should be deeply improved, taking into account published literature in the field. Differences and novelty should be highlighted in order to outline the impact of the work in the scenario of the fisetin pharmacological interest.
Author Response
In the present manuscript, the antinflammatory and antioxidant properties of the flavonoid fisetin have been evaluated in TNF-α and IL-6-activated BEAS-2B cells and in an animal model of ovalbumin induced airway inflammation. The beneficial effects of the substance have been found mediated by the modulation of different pathways involved in both inflammation and oxidative stress, including MAPK, NF-κB and Nrf2 cascades. Moreover, fisetin was able to suppress the immune response activated by inflammatory stimuli. The obtained results agree with previous published data, which highlighted a similar power of fisetin in different model of airway inflammation (see for instance Huang W, Li ML, Xia MY, Shao JY. Fisetin-treatment alleviates airway inflammation through inhbition of MyD88/NF-κB signaling pathway. Int J Mol Med. 2018 Jul;42(1):208-218. doi: 10.3892/ijmm.2018.3582; Paul, P., Majhi, S., Mitra, S., & Banerjee, E. R. (2019). Orally administered fisetin as an immuno-modulatory and therapeutic agent in a mouse model of chronic allergic airway disease. Biomedical Research and Therapy, 6(7), 3262-3273. https://doi.org/10.15419/bmrat.v6i7.553; Wu MY, Hung SK, Fu SL. Immunosuppressive effects of fisetin in ovalbumin-induced asthma through inhibition of NF-κB activity. J Agric Food Chem. 2011 Oct 12;59(19):10496-504. doi: 10.1021/jf202756f; Tanaka, T.; Takahashi, R. Flavonoids and Asthma. Nutrients 2013, 5, 2128-2143. https://doi.org/10.3390/nu5062128; Zhang H, Zheng W, Feng X, et al. Nrf2⁻ARE Signaling Acts as Master Pathway for the Cellular Antioxidant Activity of Fisetin. Molecules. 2019;24(4):708. doi:10.3390/molecules24040708). However, the true novelty of the study and its contribution to the characterization of fisetin bioactivities and pharmacological interest remain to be defined.
This represents a major limit of the study along with reading difficulties caused by a lack of clarity in the English style and contents. This last issue requires a deep and entire revision of the manuscript, likely taking advantage of the advice of English mother tongue.
Responses:
Thank for reviewer’s suggestion, we had checked the manuscript by San Francisco Edit as the supplement 1(Invoice No: 210338).
Attach a document proving English editing.
Other issues to be considered are described following.
1.Lines 78-81: The aim of the study should be better described, also explaining the study design and the choice of the applied experimental models.
Responses:
Thank for reviewer’s suggestion. We modified more detailed experimental purpose in this manuscript (line 84-87).
“Hence, we would investigate whether fisetin attenuates oxidative responses and in-flammatory cytokine expression in human tracheal epithelial cells. We also evaluate ed whether fisetin improved the molecular mechanisms underlying airway inflammatory, oxidative stress, and pathological lesions in the lungs of asthmatic mice.”
2.Cell culture studies: the choice of the concentrations used along with time exposure should be explained.
Responses:
We describe the fisetin concentrations for cell experiments in 3.1. Fisetin reduced pro-inflammatory cytokine expressions in BEAS-2B cells (Line 222-224)
“Cytotoxicity of fisetin detected using the CCK8 assay in BEAS-2B cells. Fisetin did not present significantly cytotoxic effects at a concentration ≤30 μM, and subsequent cell experiments used fisetin at 0–30 μM (Figure 1A).”
In addition, we add paragraph 2.3. BEAS-2B cell culture and fisetin treatment
“2.3. BEAS-2B cell culture and fisetin treatment (Line 106-110)
BEAS-2B were seeded into 24-well culture plates in DMEM/F12 medium. BEAS-2B cells treated with 0-30 μM fisetin at 37°C for 1 h and then incubated with 10 ng/ml TNF-α or 10 ng/ml TNF-α/IL-4 at 37°C for 24 h. The supernatants were collected and chemokine or cytokine levels detected using specific ELISA kits.”
time exposure:
In pre-test experiments, BEAS-2B cells stimulated with 10 ng/ml TNF-α for 8, 12, 16, 24, and 48hr. Activated BEAS-2B cells expressed the highest levels of IL-6 and IL-8 at 24 and 48 hours. However, at 24 and 48 hours of culture, BEAS-2B cells secreted IL-6 or IL-8 were not significantly different. Hence, BEAS-2B cells treated with 0-30 μM fisetin at 37°C for 1 h and then incubated with 10 ng/ml TNF-α or 10 ng/ml TNF-α/IL-4 at 37°C for 24 h.
3.Line 104: the number of seeded cells should be reported
Responses:
BEAS-2B cells (2 × 105 cells/ml) were used in the cell experiment of manuscript.
We added “2 × 105 cells/ml” on line 107, 112, 174,184, and 201.
4.Animal experiments: the choice of administered dose, administration route, methods for euthanasia should be clarified; moreover, equipment used to allow the reagent inhalation should be provided
Responses:
We described that mice were anesthetized and euthanized in line 150 -151.
“Mice were placed in an anesthesia box with 4% isoflurane to induce adequate an-esthesia and euthanasia.”
5.Paragraphs 2.12, 2.13, 2.14: the number of seeded cells should be reported.
Responses:
BEAS-2B cells (2 × 105 cells/ml) were used in the cell experiment of manuscript.
We added “2 × 105 cells/ml” on line 107, 112, 174,184, and 201.
6.Line 165: the choice inhibitor concentrations should be explained
Responses:
Thank for reviewer’s suggestion. We modified as “ Fisetin (10 μM) was combined with MAPK specific inhibitors, including 10μM SP600125 (JNK inhibitor), 10μM SB203580 (p38 inhibitor), and 10μM PD98059 (ERK inhibitor) (Enzo Life Sciences), to detect ICAM-1 protein level by ELISA.” (Line 207-208)
7.2.16. Data analysis: “three experiments including” and how many technical replicates?
Responses:
All results were examined based on at least three independent experiments. (Line 218-219)
8.Figures should be improved, using the same font. Font must be nonitalicized
Responses:
Thank for reviewer’s suggestion. Figures used the same font.
9.Figure captions: the number of biological and technical replicates should be reported
Responses:
We added the data presented as mean±SEM of three independent experiments, and the number of each group in Figure captions.
10.Figure 2 should be reported before the paragraph 3.2
Responses:
Thank for reviewer’s suggestion. We modified the Figure 2 before the paragraph 3.2
11.Significance should be evaluated with respect to the vehicle and the pro-inflammatory stimulus and symbols should be reported in the figures accordingly.
Responses:
Thank for reviewer’s suggestion. We added more description that inflammatory BEAS-2B cell significantly promoted the expression of inflammatory mediators compared to vehicle-treated BEAS-2B cells in the result.
12.Lines 308-309 “Thus, fisetin improved significantly oxidative stress in OVA sensitized mice”: the wording is confusing
Responses:
Thank for reviewer’s suggestion. We modified the sentence as “Thus, fisetin significantly attenuated oxidative stress in lung of OVA sensitized asthma mice.” (Line 358-359)
13.Methods and Results should be rewritten in a more clear and fluent form
Responses:
Thank for reviewer’s suggestion. We modified and rewritten Methods and Results in this manuscript.
14.Discussion should be deeply improved, taking into account published literature in the field.
Responses:
We feel great thanks for your professional review work on our article. We add more discussion about fisetin improving airway inflammation, airway hyperresponsiveness and oxidative stress in lung of asthmatic mice.
15.Differences and novelty should be highlighted in order to outline the impact of the work in the scenario of the fisetin pharmacological interest.
Responses:
We add more discussion for fisetin pharmacological characteristic in Discussion and Conclusions. Thank you again for your positive comments and valuable suggestions to improve the quality of our manuscript.

Round 2
Reviewer 1 Report
I do not have any comments.
Author Response
Comments and Suggestions for Authors
I do not have any comments.
Responses:
Thank you again for your positive comments and valuable suggestions to improve the quality of our manuscript.

Reviewer 2 Report
The Authors revised the manuscript according to the reviewer comments. However, some improvements are still required.
Particularly,
- Data discussion should be improved by considering the effects of fisetin in other models of airway inflammation, such as those described in the following papers 10.3892/ijmm.2018.3582, 10.15419/bmrat.v6i7.553, 10.1021/jf202756f, 10.3390/molecules24040708. Comparing the obtained data and the previous published can allow to highlight the novelty of the study.
- Typos and grammar inaccuracies should be checked and corrected
- The choice of antagonist concentrations is not explained
- In all the figures, the difference of activated cells (e.g. TNF-alfa stimulated) respect to ctrl should be reported, instead of calculating that of ctrl vs activated cells; moreover, significance of the treatments with respect to activated cells (e.g. TNF-alfa stimulated) can be maintained.
- The wordings “effective anti-inflammatory and oxidative effect” and “good antioxidant” should be replaced by other more appropriate. I can suggest to consider the following alternative “promising agent to counteract inflammation and oxidative stress induced ….” Or similar ones.
Author Response
The Authors revised the manuscript according to the reviewer comments. However, some improvements are still required.
Particularly,
Data discussion should be improved by considering the effects of fisetin in other models of airway inflammation, such as those described in the following papers 10.3892/ijmm.2018.3582, 10.15419/bmrat.v6i7.553, 10.1021/jf202756f, 10.3390/molecules24040708. Comparing the obtained data and the previous published can allow to highlight the novelty of the study.
Responses:
Thank for reviewer’s suggestion. Our manuscript mainly emphasizes the importance of inflamed airway epithelial cells for asthma symptoms, and also highlights that fisetin can reduce airway epithelial cell inflammation, which will contribute to reduce airway hyperresponsiveness and airway inflammation in asthmatic mice. We added more discussion in line 409-425.
Typos and grammar inaccuracies should be checked and corrected
Responses:
Thank for reviewer’s suggestion, we had checked the manuscript by San Francisco Edit as the supplement 1(Invoice No: 210338).
The choice of antagonist concentrations is not explained
Responses:
Thank for reviewer’s suggestion, we describe the choice of MAPK inhibitors concentrations in line 281-285.
In all the figures, the difference of activated cells (e.g. TNF-alfa stimulated) respect to ctrl should be reported, instead of calculating that of ctrl vs activated cells; moreover, significance of the treatments with respect to activated cells (e.g. TNF-alfa stimulated) can be maintained.
Responses:
We added and modified those description in the paragraph of Result. Thank you again for your positive comments and valuable suggestions to improve the quality of our manuscript.
The wordings “effective anti-inflammatory and oxidative effect” and “good antioxidant” should be replaced by other more appropriate. I can suggest to consider the following alternative “promising agent to counteract inflammation and oxidative stress induced ….” Or similar ones.
Responses:
Thank for reviewer’s suggestion, we carefully check sentences and use more appropriate wording in this manuscript.
